# From Early Morphometrics to Machine Learning—What Future for Cardiovascular Imaging of the Pulmonary Circulation?

**DOI:** 10.3390/diagnostics10121004

**Published:** 2020-11-25

**Authors:** Deepa Gopalan, J. Simon R. Gibbs

**Affiliations:** 1Imperial College Healthcare NHS Trust, London W12 0HS, UK; 2Imperial College London, London SW7 2AZ, UK; s.gibbs@imperial.ac.uk; 3Cambridge University Hospital, Cambridge CB2 0QQ, UK; 4National Heart & Lung Institute, Imperial College London, London SW3 6LY, UK

**Keywords:** pulmonary vascular morphometrics, pulmonary vascular imaging, pulmonary perfusion imaging, blood flow imaging, AI and pulmonary vasculature, machine learning and pulmonary circulation, deep learning and pulmonary circulation, radiomics

## Abstract

Imaging plays a cardinal role in the diagnosis and management of diseases of the pulmonary circulation. Behind the picture itself, every digital image contains a wealth of quantitative data, which are hardly analysed in current routine clinical practice and this is now being transformed by radiomics. Mathematical analyses of these data using novel techniques, such as vascular morphometry (including vascular tortuosity and vascular volumes), blood flow imaging (including quantitative lung perfusion and computational flow dynamics), and artificial intelligence, are opening a window on the complex pathophysiology and structure–function relationships of pulmonary vascular diseases. They have the potential to make dramatic alterations to how clinicians investigate the pulmonary circulation, with the consequences of more rapid diagnosis and a reduction in the need for invasive procedures in the future. Applied to multimodality imaging, they can provide new information to improve disease characterization and increase diagnostic accuracy. These new technologies may be used as sophisticated biomarkers for risk prediction modelling of prognosis and for optimising the long-term management of pulmonary circulatory diseases. These innovative techniques will require evaluation in clinical trials and may in themselves serve as successful surrogate end points in trials in the years to come.

## 1. Introduction

The pulmonary circulation is a biologically complex high-flow and low-pressure circuit that works in conjunction with the right ventricle and the various structural components of the lungs to form the cardiopulmonary unit. Although these subsystems have inherent individual characteristics, they are intractably linked and form an integrated metabolic unit that act in concert. As malfunction of one or more of these components at any spatial scale can be detrimental to the cardiopulmonary unit as a whole, it is imperative that imaging evaluation of pulmonary vascular diseases moves away from the traditional macroscopic assessment to more sophisticated methodologies that integrate structure–function relationships. The coupling of phenomenal parallel technical, engineering, and analytical advances in imaging and computer science is redefining the radiological landscape and is the catalyst for ushering in a new era of precision medicine with improved disease detection and characterisation, risk stratification and prognostication, monitoring, and directing appropriate treatment followed by assessment of response to the instituted therapy. There is a visible shift in radiological practice from the conventional model of visual observer-driven pattern recognition to the more frequent use of semi-quantitative and quantitative objective imaging biomarkers. In this article, we will discuss the evolving use of morphometric and ML tools in the imaging of the pulmonary circulation.

## 2. Pulmonary Vascular Morphometrics: Evolution from Castings to Imaging

Pulmonary morphometry is the science of understanding the quantitative anatomy of the lung, and the correlation of anatomy with physiology. The morphometric data on vascular geometry, such as the diameter, branch order and branch angles, elasticity, and connectivity matrix between the different vessels, are used to compute the total cross-sectional areas, blood volumes, and fractal dimensions in the pulmonary arteries and veins. The fundamental aim is to formulate a model for the hemodynamic circuits in order to analyse the blood flow dynamics in the lungs. Pioneering work using resin or silicone elastomer vessel casts of necropsy material led to the establishment of normal values for the number and size of pulmonary vessels in humans [1,2,3,4,5]. However, the castings-based approach to morphometry is arduous, requires a higher filling pressure compared to physiological pressures, and is not conducive to studying vascular remodelling associated with adaption to various forms of stress [6]. The lack of the surrounding microenvironment limits the applicability of the results with other measurements that might affect the whole lung.

Non-invasive imaging techniques, such as computed tomography (CT) and magnetic resonance imaging (MRI), can acquire high-fidelity images of the pulmonary vasculature with intact spatial orientation and connectivity in vivo and are a practical alternative to castings. The imaging approach also allows for rapid data collection under different experimental conditions with the added advantage of digitisation. The multiplanar three-dimensional nature of CT and MR angiography is well suited to provide quantitative morphometric information. Multidetector CT scanners are capable of acquiring sub-millimetre isotropic images of the thorax in a few seconds. A good-quality CT can depict pulmonary vessels down to diameters of about 1 mm [7], but manual vascular segmentation is prohibitively time consuming given the large number of vessels and also the separation of the arteries from veins can be challenging.

Vessel segmentation is the fundamental step in computer-aided processing of data generated by three-dimensional (3-D) imaging modalities (Figure 1). Lesage et al. [8] offers a comprehensive review of various vascular lumen segmentation models for CT and MR angiography. Highly accurate pulmonary artery–vein segmentation on CT is often a necessary prerequisite as it allows the study of both flow systems separately. This process has also evolved over the last few years and it is now feasible to use a fully automatic deep learning algorithm that utilises a 3-D convolutional neural network with minimal user interactions [9]. Advances in computational vascular morphometry can facilitate complex CT volumetric assessments of intraparenchymal vascular morphology to provide different vascular metrics that can be used to further our understanding of pulmonary vascular diseases [10]. A major advantage of automation is rapid tracking of large volumes in a shorter time frame and with reduced bias. It is also easier to standardise and hence can be applied both in routine clinical practice as well as for large imaged-based clinical studies.

Reference values for the number, volume and tortuosity of pulmonary vessels determined from morphology readouts using a fully automated vessel separation algorithm applied to computed tomography pulmonary angiography (CTPA) datasets in healthy cohorts have been published [11]. However, it is well known that there is heterogeneity in the number, diameter, branch pattern, and wall composition of the different components of pulmonary circulation that may predispose for alterations under certain pathological conditions [12].


**Clinical Applications of Morphometrics in Pulmonary Vascular Diseases**


Computer-based quantitative evaluation of the lung vessel morphology to derive imaging-based biomarkers has the potential to provide better insights into pulmonary vascular diseases (PVDs). A key pathological feature of PVD is the vascular remodelling that is characterised by cellular and structural alteration in the normal architecture of the pulmonary vessel wall. The remodelling is associated with a vicious cycle of increased pulmonary vascular resistance and intravascular pressure that eventually results in refractory right ventricular failure and death. New therapeutic approaches to PVDs require quantitative morphometric tools for correlation of vascular metrics with functional parameters. Given the heterogeneity of pathogenic factors associated with PVDs, it is necessary to choose the most relevant parameters in a specific disease setting that can then be analysed.

### 2.1. Pulmonary Hypertension: Screening, Disease Detection, Disease Severity, and a Non-Invasive Measure of Mean PAP

Imaging features that suggest vascular remodelling include pruning and loss of arterial branching [13]. Vascular tortuosity is also a frequent finding in PH. In a pilot study of 23 patients (18 with PH due to different aetiologies), Helmberger and colleagues applied a vessel enhancement filter-based automatic extraction algorithm to contrast enhanced thoracic CTs and demonstrated strong links between vascular tortuosity and mean pulmonary arterial pressure, pulmonary vascular resistance, and measurements of pulmonary gas exchange. The mean ‘distance metric’ that was used to gauge the vascular tortuosity showed a good discriminative power between patients with and without PH and hence could potentially be used in imaging-based PH screening [14]. The distance metric also demonstrated significant correlation with the WHO functional class and hence may be a reflector of disease severity.

Using automated 3-D volumetry based on CT pulmonary angiography (CTPA), a high correlation has been demonstrated between central pulmonary artery (PA) volumes and mean PA pressures in patients with and without pulmonary arterial hypertension (PAH) [15]. In a study using a regression model, the combination of main PA (MPA) volume and echocardiographically derived pulmonary artery systolic pressure (PASP) provided superior diagnostic accuracy and could have the potential for non-invasive prediction of mean pulmonary arterial pressure [16]

### 2.2. Interstitial Lung Disease (ILD): Disease Severity, Risk Stratification, and Monitoring

CALIPER (Computer-Aided Lung Informatics for Pathology Evaluation and Rating) is a software tool developed by the Biomedical Imaging Resource Laboratory at the Mayo Clinic Rochester (Rochester, MN, USA) to automatically identify and quantify changes in the lung parenchyma based on high-resolution CT images of the thorax. The software can also quantify changes to the small and large vessels by scoring them as a percentage of the volume of pulmonary vascular-related structures. The pulmonary vascular volume correlates with the extent of ILD and could therefore be a potential new index when assessing disease severity [17]. In a more recent publication, a baseline CALIPER-derived ILD extent higher than 20% and pulmonary vascular-related structures score greater than 5% was associated with worse prognosis [18]. This imaging biomarker could be a monitoring tool with an objective endpoint in future clinical trials.

### 2.3. Chronic Obstructive Pulmonary Disease (COPD): Disease Severity

Pulmonary vascular alteration is a well-recognised feature of COPD, with studies linking it to alterations in endothelial dysfunction. The development of PH is an important predictor of mortality in COPD [19]. The in vivo relationship between pulmonary arterial pressure and small-vessel morphology in emphysema was first evaluated on CT by Matsuoka et al. [20] using computer-assisted semiautomated morphometric image-processing software. The relationship assessment between the extent of emphysema and the percentage of total cross-sectional area (CSA) of sub-subsegmental pulmonary vessels less than 5 mm^2^ demonstrated that % CSA < 5 had a negative inverse correlation with mean PA pressures and emphysema severity.

The ratio of blood vessel volume in vessels <5 mm^2^ in cross-section (BV5) to total blood vessel volume (TBV) can be used as measure of vascular pruning, with lower values indicating more pruning. In smokers, the aggregate blood vessel volume in vessels less than 5 mm^2^ show the distal pruning of the intraparenchymal blood vessels and disproportionate loss of non-vascular tissue [10]. Loss of small vessel volume in CT imaging of smokers is associated with histological loss of vascular cross-sectional area [21]. Furthermore, the arterial small vessel density also shows good correlation with morphometric assessments of vessel remodelling in muscular pulmonary arteries.

### 2.4. Chronic Thromboembolic Pulmonary Hypertension: Disease Detection and Characterisation

The arterial abnormalities in chronic thromboembolic pulmonary hypertension (CTEPH) are well documented in the literature. Elegant work by Rahaghi et al. [22] provides objective evaluation of the vascular morphology on CT to facilitate lobar and global quantification. Aggregate plots of blood vessel volume versus vascular cross-sectional area on CT show distal vascular pruning and increased proximal arterial engorgement. These indices have the discriminatory power to distinguish CTEPH from the control population. CTEPH cases also showed increased arterial tortuosity and differential distribution of the volume between arterial and venous beds. These vascular metrics have the potential to be used as imaging biomarkers that can complement haemodynamic data.

### 2.5. COVID-19: Insights into the Pathophysiology of Disease

There is emergent data linking the high mortality observed among patients with severe acute respiratory syndrome coronavirus 2 (SARS-CoV-2) and pulmonary vascular involvement. Increasing attention has been placed on the concept of microvascular thrombi and dysregulated hypoxic pulmonary vasoconstriction as possible explanations for the severe hypoxemia related to COVID-19. Building upon previous work by Rahaghi et al. [22] outlined above, Lins and colleagues were able to show the value of quantitative CT analysis for the evaluation of pulmonary vascular dysfunction in 103 patients with COVID-19 [23]. Compared to healthy volunteers, COVID-19 patients showed a significant reduction in the blood volume in small vessels and an increase of the blood volume in the medium and large vessels (Figure 2). The authors postulate that this “redistribution” of blood volume within the pulmonary vascular tree could be due to increased pulmonary vascular resistance in the distal vessels that is below the resolution of CT to visualise by the naked eye. It must also be pointed out that in addition to applying an automated blood vessel segmentation algorithm, this work also used a fully convolutional deep learning model on part of the CT datasets to analyse the effect that concomitant lung parenchymal changes of COVID-19 might have on the segmentation algorithm (Lins et al.).

## 3. Pulmonary Perfusion

Assessment of lung perfusion can refine our knowledge about the pathophysiology of PVD. Contemporary imaging modalities for evaluation of perfusion are technically different and hence may provide a multitude of information that may not always be concordant. Hopkins et al. provided an excellent summary of the confounding variables in the imaging-based perfusion measurements [24,25].

### 3.1. Nuclear Medicine

West and colleagues first described the gravitational differences in pulmonary blood flow following administration of radioisotope [26]. Perfusion scintigraphy is now a mature and well-established technique for evaluation of pulmonary perfusion. Intravenous administration of ^99m^Tc-labeled macroaggregated albumin results in temporary trapping of the spheres in the pulmonary capillaries in proportion to local blood flow. Hence, scintigraphic quantification is a measure of true perfusion, as microspheres (10–150 μm) are distributed to small pulmonary arterioles and capillary beds and are not located in large conduit vessels. The spatial and contrast resolution required for defining a perfusion defect improves with the use of single photon emission computed tomography (SPECT) or SPECT/CT compared to standard planar imaging. The addition of CT allows for attenuation correction and improves the measurements of regional pulmonary perfusion, but it must be emphasised that SPECT measures of perfusion are semiquantitative relative to the overall mean perfusion. First pass pulmonary perfusion can be measured using 13N2-PET imaging, where the local concentration reflects local perfusion [27]. This has been applied in clinical practice to demonstrate spatial heterogeneity of lung perfusion in COPD and has the potential to become a vascular biomarker in airways disease [28].

Semiquantitative visual and quantitative volumetric scoring of SPECT data have numerous advances over the qualitative interpretation of ventilation-perfusion (V/Q) scintigraphy (Figure 3). Quantification may be useful for the management of patients with acute pulmonary embolism (PE) as the extent of PE is an independent risk factor for recurrence. Based on the results of VQ SPECT quantification and haemodynamic stability, a large proportion of patients with small- or medium-sized PE with a low SPECT score can be safely managed in the outpatient setting [29,30].

In patients with CTEPH, SPECT quantification can be a potential biomarker for disease severity and therapy monitoring. Derlin and colleagues demonstrated the correlation between the SPECT-derived perfusion defect score and haemodynamic parameters like mean pulmonary artery pressure (mPAP) and serum levels of N-terminal pro–B-type natriuretic peptide. Both perfusion lung volumes and perfusion index decreased with an increase in mPAP [31]. Following on from this work, a quantitative metric based on the difference in ventilation and perfusion volumes (V-P) has been used to identify discordant defects on SPECT images in CTEPH before and after treatment [32].

In PAH, perfusion is patchy due to non-homogenous obstruction of the pulmonary vascular bed. Quantification of this heterogeneity can be useful in understanding the pathophysiological sequalae of pulmonary vascular obstruction. [33]. Additionally, the perfusion index measured from planar imaging has been shown to have significant correlation with mean PAP and right ventricular ejection fraction at baseline, with improvement in the index following vasodilator therapy in PAH patients [34].

### 3.2. Dual Energy CT, DECT

The concept of dual-energy CT was first introduced in the 1970s [35], but it is only since the last decade that it has emerged as a promising tool for lung imaging. Spectral information for tissue characterisation is obtained by simultaneous acquisition of two datasets at different tube voltages to generate grey-scale images and color-coded overlays highlighting the locations of the imaging material of choice (e.g., xenon or iodine). With the use of three-material decomposition algorithms, it is possible to generate separate material-specific images [36]

DECT pulmonary angiography allows simultaneous evaluation of lung morphology, parenchymal density, and pulmonary perfused blood volumes (PBVs). Images from the iodine maps and pulmonary CT angiography (CTA) can be fused to provide morphologic and functional information from a single examination (Figure 4). Although pulmonary perfusion is a dynamic process of blood flow over time, a DECT-derived iodine map of the lung microcirculation represents blood volume measurement at one predefined point and hence is only a surrogate marker of perfusion. Nevertheless, by revealing the distribution of intravenously injected iodine contrast material in the parenchyma, PBV maps can potentially act as biomarkers of pulmonary perfusion across a wide spectrum of PVD (Figure 5).

There is a mounting body of evidence to support the complimentary functional role of DECT in the diagnosis of acute PE [37,38,39,40,41]. Subsegmental pulmonary emboli can be challenging to detect on standard CTPA-images, but the advantage of DECT is that 3–5-mm-diameter subsegmental emboli can create a 3–5-cm-diameter peripheral parenchymal perfusion defect [42] and hence the iodine maps can improve the detection rate of subsegmental emboli [38]. DE-CTPA has a sensitivity/specificity of 100% and 100% for the diagnosis of acute PE-related perfusion deficits compared to SPECT/CT and a per segment sensitivity/specificity of 83%/99% with a negative predictive value of 93% for DECT when correlated with ventilation/perfusion scintigraphy [43]. Pulmonary perfusion deficit scores are inversely correlated with thrombus load and signs of right heart strain [44]. Zhang and colleagues also found a positive relationship between PBV scores and right heart dysfunction, and advocate DECT for follow-up after institution of anticoagulation [45]. Automated quantification of pulmonary perfused blood volume has been shown to predict intensive care unit admission [46]. Thus, DECT indices have the potential for acting as a biomarker for PE severity, prognostication, and therapy monitoring.

In chronic thromboembolic pulmonary hypertension, the DE-CTPA has a high diagnostic accuracy with 100% sensitivity and 92% specificity [47]. Nakagawa and colleagues found good concordance between DECT and perfusion scintigraphy, but the work by Renapurkar et al. showed only modest inter-modality correlation in automated quantification between PBV maps and planar scintigraphy [48,49]. In patients with chronic pulmonary vascular obstruction, bronchial collaterals develop and may even participate in blood oxygenation. Whilst scintigraphy is a measure of pulmonary circulation-mediated lung perfusion, DECT PBV maps cannot distinguish the contribution from systemic circulation. Hence, it may be necessary to do dual-phase quantification as elaborated by Koike et al., where the early phase PBV reflects pulmonary arterial contribution and the late-phase PBV is the additive effect of pulmonary arterial and systemic collateral flow [50].

The extent of hypoperfusion on the colour-coded lung PBV maps may also be useful as a non-invasive estimator of disease severity [51] (Figure 6). Lung PBV scores show significant correlation with haemodynamic parameters, such as mean PAP and PVR [52,53]. Patients who have undergone successful balloon pulmonary angioplasty had positive correlation between DECT and clinical and haemodynamic parameters. Improvement in 6-min walking distance, PAP, and pulmonary vascular resistance (PVR) were associated with concomitant improvement in DECT perfusion maps [54]. DECT thus has the potential to obviate the need for recurrent right heart catheterisations in long-term follow-up of CTEPH patents as it can be used as a non-invasive monitoring tool to evaluate disease severity and treatment response.

Automated quantification of DE-CTPA-derived pulmonary PBV can be used as a reader-independent tool for the evaluation of global and regional pulmonary perfusion in emphysema. Such objective evidence of reduced pulmonary perfusion can be helpful to assess the COPD severity [55]. The role of DECT-PBV as an imaging biomarker has also been used to evaluate vascular endothelial dysfunction in smokers with emphysema. CT-derived pulmonary blood flow heterogeneity is greater in smokers with visual evidence of centriacinar emphysema (CAE) on CT even if the PFTs are normal [56]. In a study of 17 PFT-normal current smokers with and without CAE, DECT-PBV images acquired before and 1 h after administration of oral sildenafil demonstrated a statistically significant decrease in PBV-CV (coefficients of variation; a measure of spatial blood flow heterogeneity) in smokers with CAE but did not change in smokers without CAE. At baseline, the CAE group also showed higher arterial volume and cross-sectional area (CSA) in the lower lobes suggestive of arterial enlargement due to increased peripheral resistance and following sildenafil, there was a reduction in the arterial CSA. This inverse correlation between PBV change and arterial volume was not present in the non-CAE group. Thus, this work highlights the possibility of using DECT-PBV as a surrogate biomarker of reversible endothelial dysfunction [57].

### 3.3. MR Perfusion

MR perfusion techniques have been available for more than 20 years, but despite the relative ease with which it can be implemented on the modern MR platforms, pulmonary perfusion has been under-utilised outside specialist centres. However, in recent times, MR-based perfusion imaging has been gaining popularity due to emerging data regarding its widespread applicability in various forms of pulmonary vascular disease, with the added potential for reproducible quantitative assessments that can provide new insights into pathophysiological processes of the cardio-pulmonary unit.

The two main MR techniques to depict pulmonary perfusion are the non-contrast arterial spin labelling (ASL) and time-resolved dynamic contrast-enhanced (DCE) technique. ASL is not routinely used in clinical practice as it involves lengthy acquisition times and relatively low signal increase. DCE-MR imaging consists of rapid multiphasic acquisition of the first pass of gadolinium contrast media through the heart and lungs following intravenous bolus injection. The baseline pre contrast images can be subtracted from the peak intensity image to give a qualitative perfusion image. Quantitative analysis is based on the indicator dilution theory, where the maximum signal intensity and the temporal course of the signal change are used to measure tissue kinetics, such as regional pulmonary blood flow (PBF), pulmonary blood volume (PBV), and mean transit time (MTT), within the entire lung on 3-D sequences [58,59,60].

Focal areas of wedge-shaped perfusion defects allow for an indirect diagnosis of PE. In these areas of decreased perfusion, the mean PBF and PBV are decreased, and mean MTT is prolonged [61,62]. The acute PTE index, defined as the ratio between the volume of perfusion defects and the total lung volume as determined by DCE-MR, has been shown to correlate with clinical severity and has similar accuracy to the right ventricular /left ventricular (RV/LV) diameter ratio for the prediction of outcome after acute PE [63].

DCE-MR has a multifunctional role in CTEPH evaluation (Figure 7). It has a high diagnostic accuracy that is similar to perfusion scintigraphy to exclude CTEPH and thus can be used as a screening tool [64]. The pattern of perfusion defects permits distinguishing the focal defects seen in CTEPH from the diffuse perfusion reduction in PAH [65]. Dynamic perfusion MR parameters outperformed MDCT in the assessment of therapeutic response in patients with inoperable CTEPH [66]. In 20 patients with operable CTEPH, Schoenfeld and colleagues were able to confirm surgical success by demonstrating improvement in the PBV in the lower lobes and concomitant improvement in exercise capacity [67].

DCE-MR perfusion has the potential to be used for PH screening, risk stratification, prognostication, and monitoring response to treatment. Ohno and colleagues were the first to demonstrate that there are significant differences in perfusion parameters between healthy subjects and PAH patients. In PAH, there is a diffuse reduction in mean PBF and markedly prolonged MTT [68]. The first-pass clearance curve and pulmonary transit time in PAH cases correlate well with haemodynamic prognostic indictors, such as pulmonary vascular resistance and cardiac index [69,70]. Prolongation of perfusion parameters is a predictor of mortality independent of age, gender, and WHO functional class. In connective tissue disease-associated PH, there was significant correlation between PBF and PBV with mean PAP and moderate correlation with PVR [71].

The pulmonary perfusion in COPD is heterogeneously altered. Semiquantitative perfusion measures, such as PBF and PBV, are significantly reduced and MTT is prolonged. These findings show good correlation to the lung diffusing capacity [72]. In a small study of 18 patients with combined pulmonary fibrosis and emphysema (CPFE), the prolongation of the MTT and time to peak enhancement were significantly prolonged, with good correlation to haemodynamic parameters, such as mPAP and PVR index [73].

There is currently an unresolved debate regarding the safety of gadolinium contrast media [74,75], which has sparked an interest in the development of novel imaging sequences that can produce ventilation and perfusion maps without any external contrast agent. An elegant work by Schönfeld and colleagues showed perfusion-weighted Fourier-decomposition MRI to be a feasible alternative to DCE-MRI for diagnosis of chronic thromboembolic disease without resorting to ionising radiation or contrast agent [76].

Perfusion MR is a promising biomarker in the assessment of pulmonary vascular diseases. With the ongoing improvements in image acquisition techniques and standardisation of post processing software, it is likely to gain greater acceptance in routine clinical practice.

## 4. Blood Flow Imaging (BFI)

BFI refers to contemporary vascular techniques based on fluid-flow dynamics. The two main approaches to BFI are measurement-based flow visualisation using non-invasive imaging, such as echocardiography or MRI, and computational fluid dynamics (CFD) [77]. Non-invasive in vivo imaging modalities provide actual quantitative measurements without disturbing the normal biological environment. CFD is specialist area of applied mathematics that uses computer simulation programs to resolve complex issues relating to blood flow and is increasingly being used as a modelling tool to study a variety of cardiovascular conditions. Imaging techniques have limitations in temporal and spatial resolution whilst the resolution in CFD can increase the computer’s memory limit.

It is clear from the emergent publications on pulmonary circulatory CFD modelling [78,79,80,81,82] that the composite of high-performance computing, high-quality imaging data, and numerical modelling can be successfully integrated to offer insights into the complex disease mechanisms in PH. Wall shear stress (WSS) is a predictor of endothelial dysfunction and has been implicated in the transcriptional events in vascular remodelling. WSS is reduced in PH compared to healthy subjects and also exhibits a relationship with PH progression [80,81] A small proof-of-concept study in CTEPH showed the potential of CFD to quantify pulmonary artery pressure gradients, WSS, and flow topology to assist the decision-making process regarding surgical eligibility [83].

Phase-contrast magnetic resonance imaging (PC MRI) is a well-established technique for measurements of blood flow and velocity. An excellent review by Reiter et al. [84] has summarised its clinical utility in PH. Three-dimensional time-resolved multidirectional MR flow imaging (4-D-flow) provides data regarding the temporal evolution of complex flow patterns to evaluate blood flow topology, pulse-wave propagation, WSS, and energy kinetics in the pulmonary vasculature.

The flow characteristics in patients with PH are considerably different when compared with healthy and normotensive controls. In PH, there is an early onset of retrograde flow and large-scale flow vortices in the mPA (Figure 8, Appendix A). These flow parameters have a linear relationship with elevated mPAP [85,86]. Vortical blood flow in the MPA longer than 14.3% of the cardiac interval corresponds to PH with 97% sensitivity and 96% specificity [85]. A case report of a patient with CTEPH showed substantial late-systole vortex flow in the MPA with complete normalisation of the flow patterns after successful balloon pulmonary angioplasty [87].

Similar to the CFD literature, a 4-D flow MR-derived WSS also shows a significant reduction in the mPA in patients with PH [88,89] and can potentially be used as a biomarker to discriminate PH patients from healthy subjects. There is a significant negative correlation between the 4-D flow MR-derived reduction in WSS and invasive metrics, such as mPAP and PVR [90,91], as well as markers of pulmonary artery stiffness [90].

Energy loss is a parameter of cardiac workload. Han et al. showed that when compared to healthy controls, PAH patients have a significant increase in the RV kinetic energy work density and percent viscous PA energy loss [92].

Most of the published work on 4-D flow MR and its applicability in the assessment of PH is based on single-centre work on a small cohort of patients. Improving protocol standardisation coupled with the increasing availability of commercial software packages for post processing should facilitate prospective validation of the imaging-based vascular metrics in different PH groups and larger patient cohorts.

## 5. Applications of Artificial Intelligence in Multimodality Imaging of the Pulmonary Circulation and Right Ventricle

Emerging artificial intelligence (AI) approaches using biomedical imaging inputs are being used to deliver a variety of tasks ranging from triaging to workflow, disease detection and characterisation, and risk prediction modelling for prognostication and optimisation of the delivery of care to improve outcomes. Machine learning (ML) algorithms (supervised or unsupervised) are fashioned to search and extract patterns from data to provide maximum predictive ability [93,94]. Deep learning (DL) or deep convolutional neural networks (CNNs) are a subclass of ML with more processing power to perceive nonlinear structure within the data [95]. Multiscale AI modelling is apposite for non-invasive assessment of pulmonary vascular disease, with the ostensible advantage of unbiased selection.

### 5.1. Computed Tomography Pulmonary Angiography (CTPA)

CTPA is the current diagnostic imaging standard in the evaluation of suspected acute PE [96]. However, as shown in a recent retrospective study involving 8449 patients, where 99% of the scans (CTPA or V/Q) done to exclude acute PE were negative [97], there is intemperate use of imaging. This not only depletes limited resources in the healthcare system but also exposes patients to the unwarranted effects of radiation and contrast medium. To circumvent this problem, there are several risk prediction scores that can be used to make an informed decision about performing advanced imaging in the evaluation of pulmonary vascular disease. Although these have the advantage of being relatively easy to compute by clinicians at the point of care, most have not made any significant change in the imaging yield [98]. ML is capable of leveraging large volumes of complex variables to identify patterns and provide patient-specific risk scores. Objective scoring has higher discriminatory power and hence can reduce inappropriate use of imaging, as demonstrated by Banerjee et al. in the use of CTPA for the diagnosis of acute PE [99].

AI algorithms can be applied to improve parameters relating to image acquisition, contrast medium injection, and radiation dose optimisation in the acquisition of thoracic CT [100].

Computer-aided detection (CAD) for automated diagnosis of acute PE on CTPA has evolved over the last decade (Figure 9). Earlier efforts focusing on traditional feature engineering methodologies required complex pre-processing and infrastructure [101,102,103,104]. Recently, deep learning approaches have been used for PE detection and have outperformed the CAD algorithms. Using convolutional neural networks, Tajbakhsh and colleagues reported 83% sensitivity for acute PE detection with 2 false positives per volume [105]. Compared with human readers, CAD is more sensitive in the detection of peripheral emboli, particularly for inexperienced readers [104,106]. CAD algorithms can also be useful in PE risk stratification by automatic computation of the RV/LV ratio as a measure of right ventricular dysfunction [107,108].

### 5.2. Nuclear Medicine

The use of artificial neuronal networks (ANNs) to detect acute PE using ventilation-perfusion scintigraphy has been documented since the 1990s [109,110,111,112]. Holst and colleagues demonstrated the use of ANN to develop a completely automated methodology for the interpretation of V/Q scintigraphy [110]. In recent years, ML has been used for quantitative assessment of myocardial perfusion and blood flow, albeit for evaluation of coronary artery disease [113].

### 5.3. Echocardiography

Echocardiogram is often the first-line non-invasive screening tool in the work-up of patients with suspected PH. In addition to confirming PH, echocardiography can evaluate the right ventricular morphology and function and provide information on the PH aetiology, pathophysiology, and prognosis

ML can play a complementary role in enhancing the sensitivity and predictive accuracy of echocardiography for PH prediction [114]. A breakthrough work by Zhang and colleagues [115] applied a deep learning model to >14,000 echocardiograms to build a fully automated interpretation tool that can be used for disease detection and quantification of cardiac structure and function. The neural network algorithms were able to detect hypertrophic cardiomyopathy, cardiac amyloidosis, and pulmonary arterial hypertension with C statistics of 0.93, 0.87, and 0.85, respectively. Sengupta et al. demonstrated the feasibility of a cognitive ML approach to explore the multidimensional attributes of speckle tracking echocardiography in the differentiation of restrictive cardiomyopathy and constrictive pericarditis [116]. Unsupervised ML algorithms have been used in the diagnostic evaluation of patients with heart failure with preserved ejection fraction (HFpEF) [117,118]. A recent publication by Kusunose et al. underscores the possibility of using ML for automated diagnosis of regional wall motion abnormalities on echocardiography [119].

### 5.4. Cardiovascular Magnetic Resonance (CMR)

CMR is the current non-invasive imaging standard of care for assessment of RV function. DL algorithms have been used in the automatic detection and segmentation of the ventricular chambers and provide accurate quantitative measurements of RV volumes and ejection fraction [120,121] (Figure 10).

The availability of large-scale CMR data has promoted the applicability of cardiac atlases to quantify morphometric scores [122]. In a recent publication, Mauger et al. used a large cohort of 4329 CMR studies from the UK Biobank [123] to demonstrate the relationships between biventricular geometry and motion and common cardiovascular risk factors. Such AI-based characterisation of complex RV shape analysis and biventricular interactions has the potential to evaluate pre-clinical disease processes.

Geometric morphological features of the ventricles on CMR can also be used in PH prediction. Swift and colleagues demonstrated the potential of a tensor-based ML approach that allows for interrogation of CMR data without manual image segmentation [124]. This novel approach was able to differentiate patients with and without PAH with high accuracy.

In a study involving 256 patients with PH, the integration of atlas-based segmentation of the right ventricle and supervised ML of the patterns of cardiac motion was able to outperform the 4-year survival prediction based on traditional mass and volume measurements (Figure 11). The recurring ML patterns that enabled accurate outcome prediction were based on adverse structural remodelling [125]. Another ML classifier method based on MR indices has been shown to predict deterioration in patients with previous repair of tetralogy of Fallot [126]. The identification of high-risk patients may be useful for guiding surveillance frequency as well as the timing of planned intervention.

## 6. Radiomics

An elegant article by Gilles et al. explores the fundamental premise that digital images are more than pictures and their conversion to a mineable dataset is poised to become routine clinical practice [127]. Biomedical images are vast spatial data sets where every voxel is a measurement itself [128]. Radiomics is the process of extracting quantitative spatial and textural features from the images to identify potentially hidden computational biomarkers [129,130]. Such objective parameters derived using advanced bioinformatic tools can be used to identify imaging phenotypes that can be linked to apposite biologic characteristics. Radiomics is a young discipline and thus far, it has been mainly applied in the field of oncology. The analysis of the “Redefining Pulmonary Hypertension through Pulmonary Vascular Disease Phenomics (PVDOMICS)’’ multi-centre study that integrates a comprehensive omics approach with deep ML algorithms to create wide-ranging molecular profiling [131] should further our understanding of the potential place of radiomics in the large-scale panomics profiling of PVD. Radiomics enriched with other ‘omics’ data (holomics) has the potential to be used in predictive or prognostic modelling in PVD.

## 7. Extant AI-Based Pulmonary Vascular Imaging Techniques in Clinical Practice

Despite the surging popularity of AI as reflected by the exponential increase in the scientific publications in the recent years, there are numerous well-acknowledged challenges that need to be addressed. Although we are only at the beginning of a new wave of AI-fuelled enterprise, this industry is growing at an augmented rate as evidenced by the escalation in the potential applications elaborated in this article. This can be a daunting prospect to the medical profession.

Some of the AI imaging toolkits are already being used by radiologists in their current every-day practice, but this may not be readily obvious to their clinical counterparts. Examples include AI-based techniques to optimise the image acquisition process to improve image quality and reduce radiation, vascular and chamber segmentation, and extraction of quantitative features, such as calculation of ventricular function and cardiac output. These not only have the advantage of consistency and reproducibility but also have a tangible impact on the analysis time. The reporting execution is also streamlined by using AI techniques, such as automatic fetching of clinically relevant prior studies, registration and fusion of different modalities, selection of an appropriate workflow, and integration with supporting clinico-pathological information.

Some functionalities, such as CAD of acute PE, lung perfusion quantification on scintigraphy and DECT, echocardiography, and MRI-derived ventricular function analysis, and certain aspects of 4-D flow MR imaging can be readily performed using commercially available multivendor software whilst others, such as metrics for vascular remodelling, quantification of MR perfusion and morphometry, and computational flow dynamics, require proprietary software developed with in-house specialist experience and therefore are not widely available for general usage. Thus, the appositeness of some of the more complex highlighted applications is yet to be proven. Their pertinence, practicality, and economic benefits will require large-scale multicentre validation. 

Recent technological advances allow for seamless integration between smartphones and medical devices. Mobile health (mHealth) apps could potentially be valuable tools in the future to overcome some of the challenges associated with the diagnosis of diseases affecting the pulmonary circulation. Given the huge potential for mobile intervention, there is a compelling need to develop high-quality apps that will support AI-based model-informed precision diagnosis.

The lack of well-annotated public databases and the associated predicament of validation, governance issues regarding patient-sensitive data, and regulatory restrictions need to be resolved before the full potential of AI technology can be implemented in routine clinical practice. 

## 8. Conclusions

Imaging has a crucial and increasing role to play in the management of pulmonary vascular disease. Imaging-based computing of pulmonary vascular physiology can enhance diagnosis and long-term monitoring, reduce the need for invasive instrumentation, and also provide metrics that are not visible to the naked eye. The inherently quantitative nature of multidimensional pulmonary vascular morphometrics has the potential to be used as an imaging biomarker that can objectively inform on vascular remodelling. Multimodality imaging of regional pulmonary perfusion facilitates capture of the spatial and temporal heterogeneity of the pulmonary circulation. The efficacy of novel blood flow imaging techniques and computational fluid dynamics must be demonstrated in multicentre clinical trials, but it is clear that these techniques have the potential to change clinical practice. An auxiliary diagnostic system has evolved with the exponential growth of sophisticated AI-based applications in cardio-pulmonary imaging. Whilst it is important that both clinicians and radiologists embrace the opportunities provided by AI, the challenges in incorporating these algorithms into routine clinical workflow should not be underestimated.

## Figures and Tables

**Figure 1 diagnostics-10-01004-f001:**
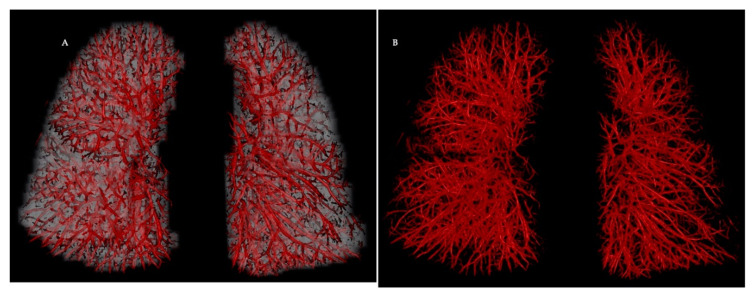
Representative computed tomography pulmonary angiography images in the coronal plane of a normal male subject with automated 3-D rendering of the intraparenchymal vasculature. The reconstruction with (panel **A**) and without lung overlay (panel **B**) demonstrates the ability of a good quality CTPA to depict subsegmental vessels. Volumetric reconstructions can be color-coded based on vessel radii and automated vessel segmentation can be used to create a blood volume distribution profile by measuring the vascular cross-sectional area in regions of interest. Image courtesy of Dr Wei Liao, Imaging Solutions, Bayer AG Engineering & Technology, Computer Vision Innovation.

**Figure 2 diagnostics-10-01004-f002:**
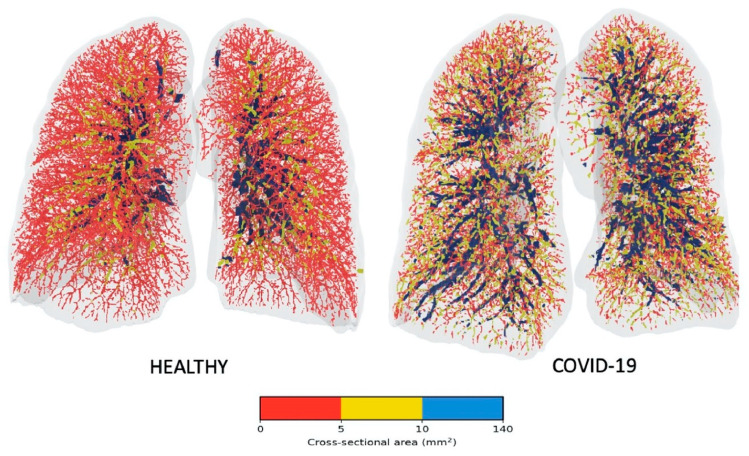
Visual representation of the blood vessels coloured according to their size. Red denotes the small vessels, yellow the mid-size vessels, and blue the larger vessels. *Reprinted from ‘Academic Radiology’, Volume 27, Issue 10, pp. 1449–1455. Lins M, Vandevenne J, Thillai M, Lavon BR, Lanclus M, Bonte S, Godon R, Kendall I, De Backer J, De Backer W. Assessment of Small Pulmonary Blood Vessels in COVID-19 Patients Using HRCT. Figure 4, Page 1453. Copyright (^©^ 2020 The Association of University Radiologists), with permission from Elsevier.*

**Figure 3 diagnostics-10-01004-f003:**
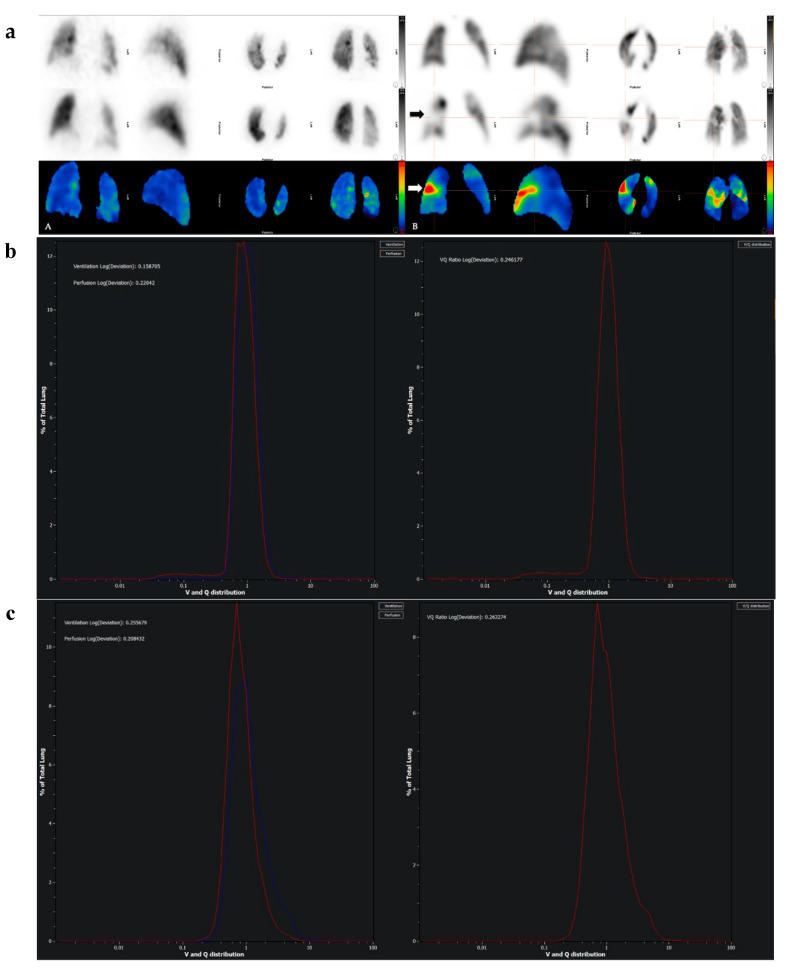
Lung VQ SPECT quantitative analysis and display from Hermes Medical Solutions’ Hybrid Viewer software. Objective analysis of SPECT scintigraphy has a high diagnostic accuracy in patients with suspected pulmonary embolism. (**a**): Panel A: Normal. Panel B: Acute pulmonary embolism (block arrows); Top row = ventilation SPECT reconstruction; middle row = perfusion SPECT reconstruction, corrected for counts remaining from the ventilation study by subtraction; bottom row = 3-D V/Q ratio image created by dividing the ventilation counts by the perfusion counts for each voxel, displayed on a 0–6 scale. In the bottom row, red represents high V/Q values, indicating a mismatch, blue represents low V/Q values, indicating that ventilation and perfusion are matched in that region. (**b**) Normal and (**c**) Acute pulmonary embolism (same patients from (**a**)). Histograms of the V and Q voxel-wise frequency distributions (left hand side) and V/Q voxel-wise frequency distribution (right hand side) plotted against the V/Q ratio. The wider the peaks and the more different the V and Q frequency distribution curves, the higher the probability of PE.

**Figure 4 diagnostics-10-01004-f004:**
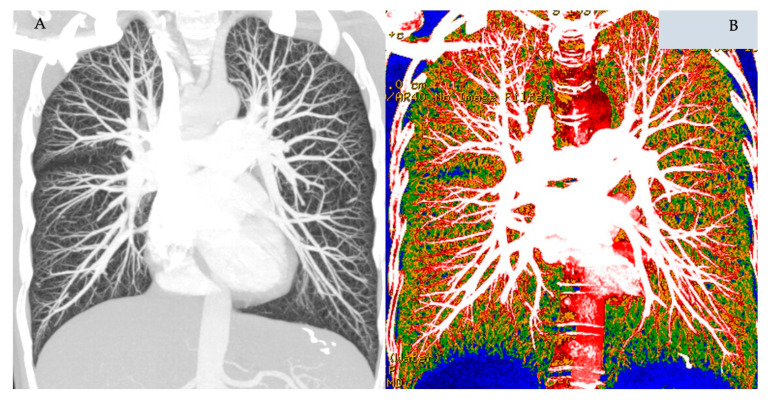
Coronal view of a dual-energy CT pulmonary angiography dataset from a normal female subject. Panel **A** is maximum intensity projection view. Panel **B** is colour-coded dual-energy overlay map.

**Figure 5 diagnostics-10-01004-f005:**
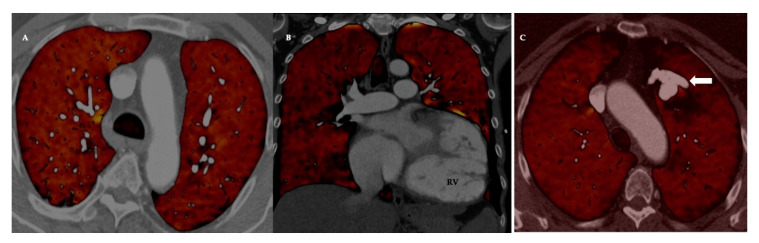
Dual-energy CT pulmonary angiography with normal perfusion (Panel **A**), heterogenous mottled perfusion in pulmonary arterial hypertension (Panel **B**), and regional reduction in perfusion around a pulmonary arteriovenous malformation (Block arrow, Panel **C**). Note the dilated and hypertrophied right ventricle (RV) in the middle panel.

**Figure 6 diagnostics-10-01004-f006:**
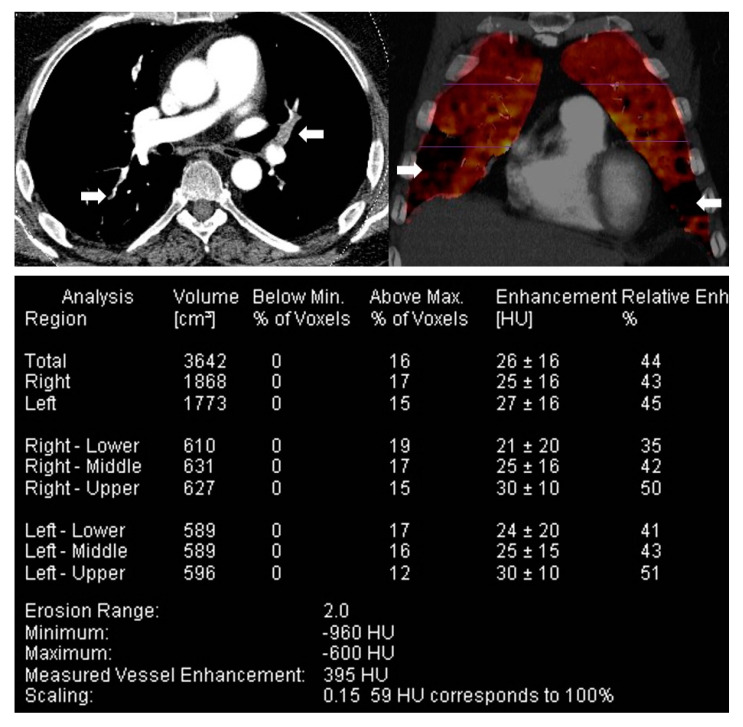
Dual-energy CT in acute pulmonary embolism (PE).Top row: Panel A: Axial CT pulmonary angiography with multiple acute PE (block arrows). Panel B: Corresponding coronal colour-coded overlay of a pulmonary perfusion map with multifocal wedge-shaped perfusion defects (block arrows). Bottom row: The numerical information corresponding to the perfused blood volume shows the lung volume (cm³) and iodine perfusion (enhancement in Hounsfield Unit, HU). The right and left lobes of the lung are visualised and evaluated independently. Both lobes of the isolated lung are subdivided into upper, middle, and lower parts based on the volume of the segmented lobes. Image courtesy of Dr Rahul D. Renapurkar, Departments of Thoracic and Cardiovascular Imaging, Imaging Institute, Cleveland Clinic.

**Figure 7 diagnostics-10-01004-f007:**
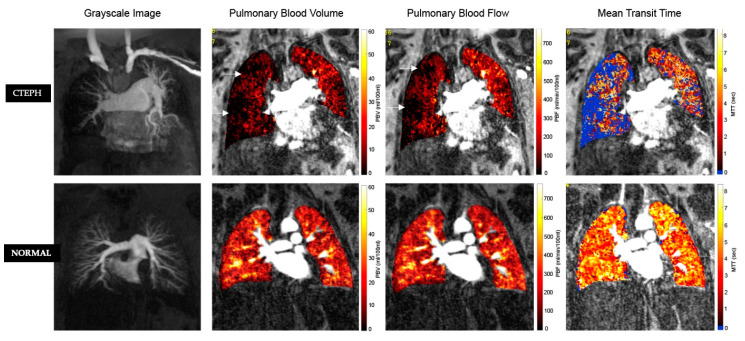
MR perfusion with absolute quantification of perfusion parameters. Top row: CTEPH. Bottom Row: Normal. From Left to Right: Maximum Intensity Projection (MIP) of MR pulmonary angiography, Pulmonary Blood Volume (PBV), Pulmonary Blood Flow (PBF), and Mean Transit Time (MTT). In chronic thromboembolic disease, there are focal areas (arrows) of wedge-shaped perfusion defects. In these areas of decreased perfusion, the mean PBF and PBV are decreased, and mean MTT is prolonged (colour maps).

**Figure 8 diagnostics-10-01004-f008:**
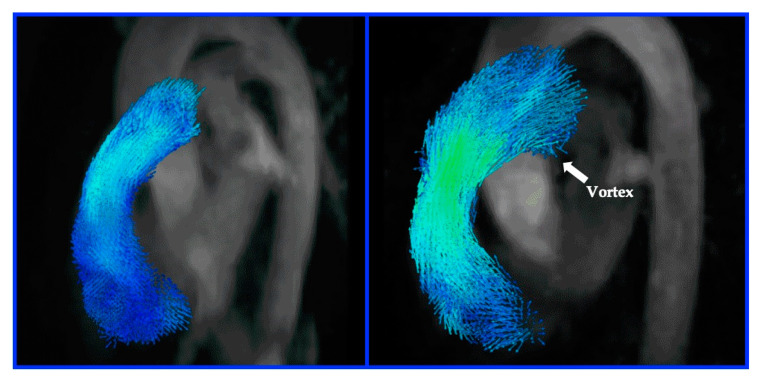
Also has corresponding Video (avi file 1). 4-D flow MR. Left panel: Normal flow. Right panel: Abnormal flow vortex (block arrow) in a patient with PH. There is a linear relationship between t_vortex_ and mean pulmonary artery pressure (mPA) that allows for diagnosis of PH. Images courtesy of Dr Ursula Rieter, Department of Radiology, Medical University of Graz and Dr Gert Rieter, Research & Development, Siemens Healthineers, Graz, Austria.

**Figure 9 diagnostics-10-01004-f009:**
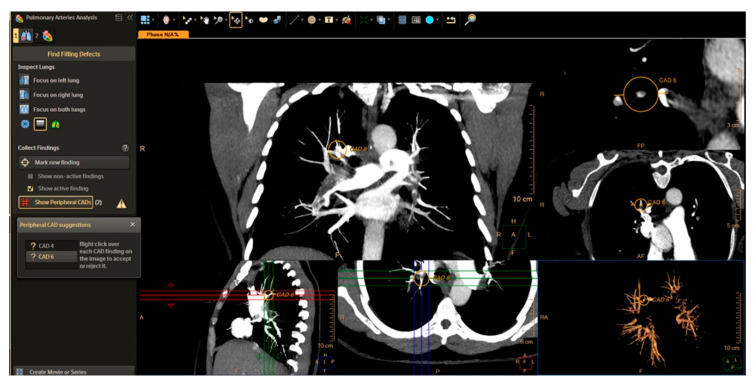
Computer-aided detection (CAD) analysis and display from Philips Healthcare, Best, The Netherlands. CT pulmonary angiographic 3-D data is displayed in three orthogonal views. A CAD algorithm (an architecture of computer image analysis process) has been applied to this CTPA, resulting in a yield of automated prompting (orange overlay) of foci suggestive of intraluminal pulmonary arterial filling defects.

**Figure 10 diagnostics-10-01004-f010:**
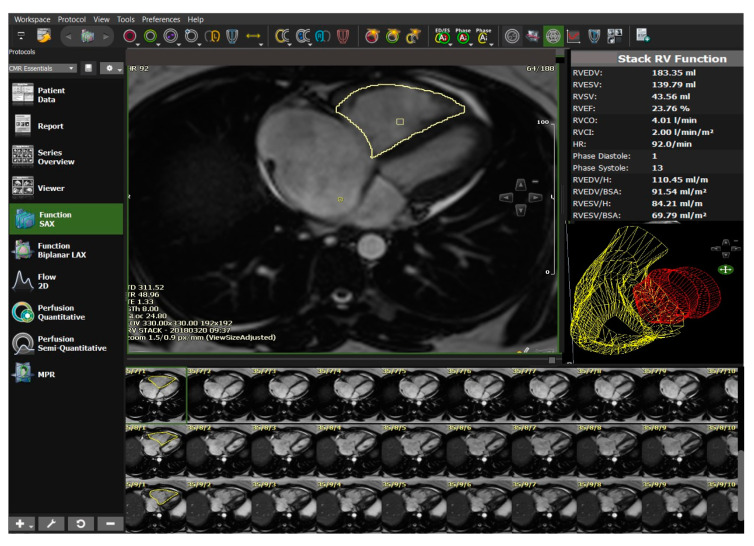
Automated right ventricular quantitative analysis and display from cvi42 MR-Circle Cardiovascular Imaging Viewer software. In this 37-year-old female with idiopathic pulmonary arterial hypertension, there is significant impairment of RV systolic function with RV ejection fraction of 24%. Automatic contour detection based on deep learning algorithms allows for a reduction in contouring times to seconds and has been shown to have an accuracy equivalent to human performance with consistent reproducibility.

**Figure 11 diagnostics-10-01004-f011:**
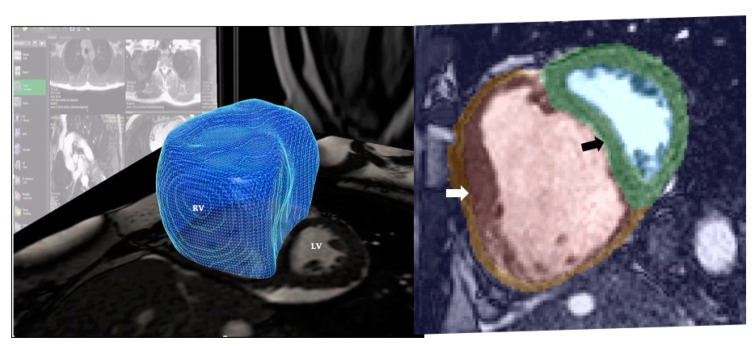
Example of computational modelling for a patient with idiopathic pulmonary arterial hypertension. A 3-D model at end-diastole and end-systole is used to determine the direction and magnitude of systolic excursion at each corresponding anatomic point in the mesh by using a deformable motion model. A machine learning algorithm was applied to identify recurring patterns that enabled survival prediction. A reduction in both longitudinal basal motion and decrease in radial contraction in the septum and free wall were found to be associated with poor outcome. RV: Right Ventricle. LV: Left Ventricle. White Arrow: Marked RV hypertrophy. Black Arrow: Abnormal motion of interventricular septum. Images courtesy of Dr Timothy Dawes, Honorary Clinical Lecturer in the National Heart & Lung Institute, London, UK.

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
