# Peer review of "From Early Morphometrics to Machine Learning—What Future for Cardiovascular Imaging of the Pulmonary Circulation?"

_diagnostics, 2020, doi:10.3390/diagnostics10121004_

Round 1

Reviewer 1 Report

Journal: Diagnostics (ISSN 2075-4418)

Manuscript ID: diagnostics-994154

Title: From early morphometrics to machine learning - what future for cardiovascular imaging of the pulmonary circulation?

Authors: Deepa Gopalan, J. Simon R. Gibbs

Overview and general recommendation:

In recent years, medical imaging plays a significant role in diagnosis and disease management. Use of morphometric and machine learning tools in the cardiovascular imaging of the pulmonary circulation have the potential to make dramatic alterations to how clinicians investigate the pulmonary vascular diseases with rapid diagnosis and minimizing the need for invasive procedures.  This review article explains the visible shift in the current radiological practice from the conventional model of visual observer-driven pattern recognition approach to modern use of quantitative imaging biomarkers in management of pulmonary circulatory diseases. It presents the results of original research and makes a valuable contribution to knowledge and understanding of novel techniques applied to multimodality imaging, which provides new information to improve disease characterization and increase diagnostic accuracy. Because of this, the current study is on a topic of relevance and general interest to the readers of the journal ‘Diagnostics’.

Overall, the paper is well-written, well-organized and well-illustrated.   I recommend the following suggestion to be done to the paper.

Minor Comments:

Mobile health can be included as one of the future scopes of AI and machine learning. Example reference “Marlon F. Alcantara et al., “Improving tuberculosis diagnostics using deep learning and mobile health technologies among resource-poor communities in Perú”, Smart Health, Volumes 1–2,2017, Pages 66-76, ISSN 2352-6483, https://doi.org/10.1016/j.smhl.2017.04.003.

Author Response

Response: Smartphone integration technology including mobile phone applications (apps), and telemedicine is increasingly considered as a way to improve access to prevention, diagnosis and management of many chronic diseases. Although some of the apps may seem promising, many contain errors or provide harmful or wrong information. Hence whilst we agree that such mobile health (mHealth) apps could potentially be a valuable tool, their clinical benefits for PVD diagnosis remains to be proven. We have added the following paragraph to Section 7 [Extant AI based Pulmonary Vascular Imaging Techniques in Clinical Practice]

Recent technological advances allow for seamless integration between smartphones and medical devices. Mobile health (mHealth) apps could potentially be valuable tools in the future to overcome some of the challenges associated with the diagnosis and management of diseases affecting the pulmonary circulation. Given the huge potential for mobile intervention, there is a compelling need to develop high quality apps that will support AI based model-informed precision diagnosis. 

Reviewer 2 Report

General comments:

The authors well reviewed and discussed the imaging modalities of vascular disease in pulmonary circulation from the early morphometrics to machine learning. Their article is likely to help readers to learn the current perspectives in the imaging modalities of vascular disease in pulmonary circulation. Although the review for each section has been adequately addressed, several changes are required to update the manuscript.

Specific comments:

Major:

#1. Vascular endothelial dysfunction is inferred to be associated with impaired vasodilation, augmented vasoconstriction and vascular remodeling. The authors demonstrated the role of DECT-PBV as an imaging biomarker of vascular endothelial dysfunction in smokers with COPD without any interpretation (page 10, lines 308-309). The authors should explain how the findings of DECT-PBV revealed the vascular endothelial dysfunction in smokers with COPD.

Minor:

#1. Descriptions of “pulmonary hypertension” and “machine learning” should be unified into their abbreviations throughout the text because the abbreviations already have been described.

Author Response

Major:

#1. Vascular endothelial dysfunction is inferred to be associated with impaired vasodilation, augmented vasoconstriction and vascular remodeling. The authors demonstrated the role of DECT-PBV as an imaging biomarker of vascular endothelial dysfunction in smokers with COPD without any interpretation (page 10, lines 308-309). The authors should explain how the findings of DECT-PBV revealed the vascular endothelial dysfunction in smokers with COPD.

Response: The following explanation, one new reference and 3 new abbreviations have been added.

CT derived pulmonary blood flow heterogeneity is greater in smokers with visual evidence of CAE on CT even if the PFT’s are normal [New Ref].  In a study of 17 PFT-normal current smokers with and without CAE, DECT-PBV images acquired before and 1 hour after administration of oral sildenafil demonstrated a statistically significant decrease in PBV-CV (coefficients of variation; a measure of spatial blood flow heterogeneity) in smokers with CAE but did not change in smokers without CAE. At baseline, the CAE group also showed higher arterial volume and CSA in the lower lobes suggestive of arterial enlargement due to increased peripheral resistance and following sildenafil, there was a reduction in the arterial CSA. This inverse correlation between PBV change and arterial volume was not present in the non-CAE group. Thus, this work highlights the possibility of using DECT-PBV as a surrogate biomarker of reversible endothelial dysfunction.

New Ref: Alford SK, van Beek EJ, McLennan G, Hoffman EA. Heterogeneity of pulmonary perfusion as a mechanistic image-based phenotype in emphysema susceptible smokers. Proc Natl Acad Sci USA 2010;107:7485–7490

New abbreviations: centriacinar emphysema (CAE), pulmonary function tests (PFT), cross sectional area (CAE)

Minor:

#1. Descriptions of “pulmonary hypertension” and “machine learning” should be unified into their abbreviations throughout the text because the abbreviations already have been described.

Response: We have now used the unified abbreviation of  ‘pulmonary hypertension’ to PH and ‘Machine Learning’ to ML where appropriate within the main body of the article as well as in the figure legends.